# An Unenhanced Breast MRI Protocol Based on Diffusion-Weighted Imaging: A Retrospective Single-Center Study on High-Risk Population for Breast Cancer

**DOI:** 10.3390/diagnostics13121996

**Published:** 2023-06-07

**Authors:** Anna Rotili, Filippo Pesapane, Giulia Signorelli, Silvia Penco, Luca Nicosia, Anna Bozzini, Lorenza Meneghetti, Cristina Zanzottera, Sara Mannucci, Bernardo Bonanni, Enrico Cassano

**Affiliations:** 1Breast Imaging Division, IEO European Institute of Oncology IRCCS, 20141 Milan, Italy; 2Division of Cancer Prevention and Genetics, IEO European Institute of Oncology IRCCS, 20141 Milan, Italy

**Keywords:** breast neoplasm, radiology, magnetic resonance imaging, diffusion weighted imaging, prevention, personalized medicine

## Abstract

Purpose: This study aimed to investigate the use of contrast-free magnetic resonance imaging (MRI) as an innovative screening method for detecting breast cancer in high-risk asymptomatic women. Specifically, the researchers evaluated the diagnostic performance of diffusion-weighted imaging (DWI) in this population. Methods: MR images from asymptomatic women, carriers of a germline mutation in either the BRCA1 or BRCA2 gene, collected in a single center from January 2019 to December 2021 were retrospectively evaluated. A radiologist with experience in breast imaging (R1) and a radiology resident (R2) independently evaluated DWI/ADC maps and, in case of doubts, T2-WI. The standard of reference was the pathological diagnosis through biopsy or surgery, or ≥1 year of clinical and radiological follow-up. Diagnostic performances were calculated for both readers with a 95% confidence interval (CI). The agreement was assessed using Cohen’s kappa (κ) statistics. Results: Out of 313 women, 145 women were included (49.5 ± 12 years), totaling 344 breast MRIs with DWI/ADC maps. The per-exam cancer prevalence was 11/344 (3.2%). The sensitivity was 8/11 (73%; 95% CI: 46–99%) for R1 and 7/11 (64%; 95% CI: 35–92%) for R2. The specificity was 301/333 (90%; 95% CI: 87–94%) for both readers. The diagnostic accuracy was 90% for both readers. R1 recalled 40/344 exams (11.6%) and R2 recalled 39/344 exams (11.3%). Inter-reader reproducibility between readers was in moderate agreement (κ = 0.43). Conclusions: In female carriers of a BRCA1/2 mutation, breast DWI supplemented with T2-WI allowed breast cancer detection with high sensitivity and specificity by a radiologist with extensive experience in breast imaging, which is comparable to other screening tests. The findings suggest that DWI and T2-WI have the potential to serve as a stand-alone method for unenhanced breast MRI screening in a selected population, opening up new perspectives for prospective trials.

## 1. Introduction

Breast cancer is increasingly a global cause for concern, owing to its high incidence around the world, and it is one of the leading causes of cancer death in females [1]. Germline mutations in several genes linked to DNA repair have been shown to be associated with an inherited risk for breast cancer [2]. Inherited mutations in BRCA1 and BRCA2 predispose individuals to high risks of breast and ovarian cancers, with lifetime risks of breast cancer as high as 80% in the US [2]. Carriers of germline BRCA exhibit tumors with peculiar pathologic features [3], such as a lack of estrogen receptor (ER), progesterone receptor (PR), and human epidermal growth factor receptor 2 (HER2) expression, displaying the “triple negative” (TN) phenotype [4].

However, mammographic screening strategies, tailored to the biology and demographics of the more common and less lethal luminal cancers, are not suitable for screening these patients.

Magnetic resonance imaging (MRI) represents the most sensitive technique to detect breast cancer, and this is recommended annually in addition to mammography for screening women with an inherited BRCA1 or BRCA2 mutation [5,6]. Despite the clear benefits of breast MRI in cancer screening, the high cost, patient tolerance, and expertise remain key issues. Abbreviated breast MRI protocols have recently emerged as an alternative to standard breast MRI protocols. These abbreviated protocols seek to reduce acquisition time by maintaining only a selected number of sequences to yield faster overall imaging times and expand patient tolerance and access [7,8,9]. Several different abbreviated protocols have already been tested, providing diagnostic performance equal to that of full breast MRI protocols [7,10]. Since all these abbreviated protocols rely on contrast administration, the relatively recent discovery that gadolinium accumulates in the brain of patients with normal renal function has raised concern about the safety of contrast-enhanced MRI [11,12].

Diffusion-weighted imaging (DWI) is increasingly being implemented into routine breast MRI protocols by the breast-imaging community, and breast DWI indications range from lesion detection and the differentiation of benign and malignant lesions to the assessment and prediction of responses to neoadjuvant chemotherapy [13]. DWI has the potential to serve as a contrast-free MR screening method. The major advantage of diffusion MRI is its ability to display tumor cellularity and microstructure at the cellular level, without the use of contrast agents, but spoiling the diffusion of water molecules in biological tissues [14]. Such a promising approach, referred to as an unenhanced protocol, cuts contrast injection and has the potential to provide information on the cellular organization of tissues, thereby shortening the examination time. Retrospective studies have shown unenhanced breast MRI to be equal to standard full protocol regarding the diagnostic accuracy [15]. However, there is no currently available data considering DWI as a screening tool for high-risk patients.

Accordingly, the motivation behind this study is to investigate the potential of a contrast-free MRI technique using DWI supplemented with T2-WI for breast cancer screening in women with a high risk of developing breast cancer due to a germline mutation in either the BRCA1 or BRCA2 gene, aiming to open new perspectives for prospective trials investigating the potential role of unenhanced breast MRI for screening in a selected population, potentially leading to improving early detection and outcomes with a more cost-effective screening method for women with a high risk of developing breast cancer.

In this retrospective study, we review DWI images of asymptomatic high-risk women that undergo MRI with a screening indication to determine its accuracy in breast cancer detection and to assess lesion visibility of clinically significant cancer.

## 2. Material and Methods

### 2.1. Patient Population

Our population included asymptomatic women with BRCA1/2 mutations who underwent breast MRIs, namely, full protocol MRI (FP-MRI), from January 2019 to December 2021 in our Institute.

Institutional Review Board approval was obtained for this retrospective study (UID 3033), and all participants signed a written informed consent before performing breast MRI.

All patients were over the age of 18 years, not pregnant or breastfeeding, and had no contraindications to MRI.

Table 1 shows inclusion and exclusion criteria. Particularly, exclusion criteria included the absence of patient follow-up or final pathological results, and patients who underwent neoadjuvant therapy because the size of locally advanced or relatively large lesions could represent a potential bias in our agreement assessment. As the FP-MRI is meant for asymptomatic patients, the presence of breast cancer and/or symptoms or signs of breast cancer or recurrence were criteria of exclusion. Moreover, we excluded patients with bilateral breast implants, as they usually cause artifacts on DWI [16]. Finally, contraindications to MRI include the presence of non-MRI-conditional implants or devices, and unmanageable claustrophobia.

Standards of reference were the pathological analysis through biopsy or surgery or ≥1 year of clinical and radiological follow-up.

### 2.2. MRI Technique

The MR examinations were performed with the patient in a prone position using a 1.5 T scanner (Optima MR450w^®^, GE Healthcare, Milwaukee, WI, USA) equipped with a 34 mT/m gradient and a dedicated 8-channel breast coil. The FP-MRI in our Institute included a 3-plane localizer, axial FSE T2-weighted images (T2-WIs), axial-diffusion-weighted images (DWIs) with the relative apparent diffusion coefficient (ADC) maps, dynamic series performed before and 4 times after intravenous administration of 0.1 mmol/kg of a gadolinium chelate at 90 s, post-processing subtraction, and maximal intensity projection (MIP) images. The T1-weighted dynamic series were not taken into consideration for the purpose of this study.

The technical parameters of the two-dimensional echo-planar spin-echo DWI sequence changed during our evaluation period. They were optimized in mid-2020 since the Institute adopted the minimum requirements for breast DWI according to the publication of the European Society of Breast Imaging (EUSOBI) guidelines [13]. The technical parameters for breast DWI are shown in Table 2. The DWI acquisition time was approximately 4 min (mean time: 4:57 min, range: 4:02–6:22 min), depending mainly on breast size, while the standard protocol described above required approximately 22 min.

### 2.3. Image Analysis and Readers’ Characteristics

A breast imaging radiologist with 15 years (R1) of experience and a radiology resident with 10 months of experience in breast MRI (R2) independently assessed DWI, including low- and high-b-value images and the ADC map. The ADC values were reported at reader’s discretion for target regions of interest (ROI), but no specific ADC thresholds were defined to consider a lesion as malignant. The two readers were blinded to medical history, imaging reports, and dynamic study, including post-processed images, such as subtracted images or MIP. In case of doubts, readers could read the T2-WI.

Each reader classified breast density into four subcategories according to BI-RADS lexicon of the American College of Radiology (ACR) [17]: ACR A (“almost entirely fatty”), ACR B (“scattered areas of fibroglandular density”), ACR C (“heterogeneously dense breasts”), and ACR D (“extremely dense breasts”).

Since a screening-like reading should be adopted on a per-exam assessment, the scale was dichotomized into two categories evaluated for the presence of cancer: negative exam, similar to “no recall”, versus positive/suspect exam, similar to “recall for further assessment” in the screening setup. Diagnostic criteria for “recall” were hyperintensity on images acquired with b = 800 s/mm^2^ with corresponding hypointensity on the ADC map, associated with the subsequent morphological evaluation of the corresponding findings on the T2-weighted image. 

In recall cases, both readers reported the localization and the diameter of the largest suspicious finding to ensure the identification of the same target lesion, and the ADC value was obtained by drawing an ROI completely within the lesion on the ADC map.

The exams were evaluated and scored based on the BI-RADS diagnostic classification. However, in the final unenhanced assessment, the study did not permit the use of the BI-RADS 0 category, and BI-RADS 6 was not possible due to blinded reading, which was intended to mimic a screening scenario. As a result, the scale was divided into two categories: negative (BI-RADS 1, 2) and positive (BI-RADS 3, 4, and 5), consistent with standard screening readings. The recall examinations involved measuring the largest diameter of the main lesion.

Interpretation time for DWI images was recorded for both R1 and R2 on a randomly selected sample of 10 patients.

### 2.4. Statistical Analysis

Per-exam sensitivity, specificity, and accuracy, and positive and negative predictive values were calculated for R1 and R2 single readings. Point estimates were given with a 95% confidence interval (CI), and descriptive statistics were reported as mean ± standard deviation (SD) or median and interquartile range (IQR) according to normal/near-normal or non-normal data distribution. Statistical calculations were performed using Microsoft Office Excel 2022, Microsoft Corporation. (2018), retrieved from https://office.microsoft.com/excel accessed on 28 February 2023.

The agreement was assessed through the calculation of inter-reader reproducibility using Cohen’s kappa (κ) statistics using R software version 4.0, retrivied from https://www.r-project.org/ accessed on 28 February 2023. The values of κ were considered as follows: 0–0.20, slight agreement; 0.21–0.40, fair agreement; 0.41–0.60, moderate agreement; 0.61–0.80, substantial agreement; 0.81–1, almost perfect agreement [18].

## 3. Results

Of the 313 women with known BRCA1/2 mutations who underwent FP-MRI, 145 met our inclusion criteria (Table 1), and they were selected in our study as reported in the flowchart (Figure 1). Finally, the two readers evaluated 344 out of 349 FP-MRI (5 FP-MRI were excluded due to the lack of DWI).

Women included in our study were aged 49.5 ± 12 years (mean ± SD; range, 20–80 years).

Many excluded women missed their FP-MRI due to the COVID-19 pandemic [19,20] or because they underwent bilateral prophylactic mastectomy.

Per-exam cancer prevalence was 11/344 (3.2%). The features of breast cancers are shown in Table 3. The median lesion size was 9 mm (IQR, 20–4 mm). Follow-up ranged from 12 to 24 months.

Regarding breast density, we reported 52/344 (15.1%) ACR A, 96/344 (29.9%) ACR B, 139/344 (40.4%) ACR C, and 57/344 (16.6%) ACR D.

We reported 186/344 (54.1%) BI-RADS 1, 131/344 (38.1%) BI-RADS 2, 15/344 (4.3%) BI-RADS 3, and 12/344 (3.5%) BI-RADS 4.

R1 recalled 40/344 exams (11.6%) and R2 recalled 39/344 exams (11.3%). The per-exam diagnostic performance of breast DWI is reported in Table 4. The sensitivity was 8/11 (73%; 95% CI: 46–99%) for R1 and 7/11 (64%; 95% CI: 35–92%) for R2. The specificity was 301/333 (90%; 95% CI: 87–94%) for both readers. The diagnostic accuracy was 90% for both readers.

Inter-reader reproducibility between readers was in moderate agreement (κ = 0.43).

## 4. Discussion

In this study, we found that the unenhanced DWI protocol could be a viable alternative for screening asymptomatic female BRCA1 and 2 mutation carriers, with a sensitivity of 73% and specificity of 90%. The inter-reader agreement was moderate.

Therefore, although a DWI-based protocol is still inferior in lesion detection than a contrast-agent-based MRI protocol, it may be considered a valid alternative in patients unsuitable for contrast agent administration or in those subgroups of patients who routinely undergo the use of contrast agents.

Moreover, BRCA1/2 carriers are recommended to undergo yearly mammography and breast MRI, alternating every six months. Considering the good performance of DWI, we might reframe the surveillance of high-risk women by adding DWI as a better alternative to mammography or ultrasound for women with high breast density [21].

Breast MRI has been extensively demonstrated to be the most powerful tool for breast imaging [22,23], even though it remains underutilized largely due to concerns about cost. One contributing factor is the long acquisition and table times to perform a standard full contrast-enhanced protocol, which includes fat-saturated T2 and dynamic contrast-enhanced T1-weighted images and ranges between 20 and 30 min. These sequences are still required by the ACR for accreditation of breast MRI and are still considered essential for the reliable detection and characterization of breast lesions [24,25].

Screening MRI among high-risk patients is recommended annually by multiple guidelines, as its use has been shown to result in higher detection rates and an earlier stage of disease at diagnosis [26,27,28]. Nevertheless, in high-risk patients, we need to consider a uniquely important issue by the routine use of a contrast-enhanced MRI protocol, such as the repeated administration of gadolinium-based contrast agents [24,29,30].

For the aforementioned reasons, there is an urgent need to develop a safe and cost-effective breast MRI protocol in order to expand its use, without compromising diagnostic performance or decreasing reader reproducibility [31,32]. Our study is a further step forward in the hot debate of whether DWI-based imaging can be a viable alternative to contrast-based MRI protocols or, at least, be an alternative for patients unsuitable for contrast agent application.

In 2014, Kuhl CK et al. introduced the concept of an abbreviated breast MRI protocol (AB-MRI), which was then supported by several studies showing comparable results with DCE MRI [7,33,34]. However, this novel approach still requires intravenous administration of contrast agents, keeping it time-consuming, costly, painful, prone to complications, and unsuitable for several patients [35]. In addition, the controversy about the safety of GBCAs has sparked the recommendation to use them only when essential diagnostic information cannot be obtained with unenhanced scans [36].

In the field of breast cancer screening, the contrast issue is particularly relevant in high-risk patients because these women are annually screened with contrast-MRI, undergoing an overload of it. Currently, DWI is revolutionizing the field of breast MRI, as it can serve as an alternative to contrast-agent-enhanced sequences. Even if, based on a different principle than tumor angiogenesis, DWI performs well in different clinical scenarios [37,38,39]. DWI sequences, based on the random Brownian motion of water molecules within a tissue, provide functional information on tissue microstructure without contrast administration. Breast cancers present an increase in cell density, hindering water molecule motion, showing a higher signal on DWI and a lower signal on ADC map, which is in contrast to benign lesions and normal tissue [40]. Currently, DWI has already been included in multiparametric FP-MRI, reducing the high false positive rate of DCE MRI [41]. Several authors have investigated abbreviated unenhanced protocols with different combinations of T1-weighted and/or T2-weighted images with different b values, with encouraging results [15,42,43,44,45].

In this retrospective study, we tested the diagnostic performance of DWI sequences, optionally supplemented with T2-weighted sequences, as a tool for cancer detection in asymptomatic BRCA carriers undergoing annual MRI screening. Our encouraging results pave the way to reframe the surveillance of these women. In addition, breast cancers arising in mutation carriers tend to exhibit adverse histopathologic features that are indicative of aggressive biologic behavior, they exhibit high proliferation rates, are more likely to show high nuclear grading, and are more often receptor-negative. We analyzed the histotype of our false negative cases, finding that only one was a biologically relevant cancer. The aim of a screening campaign should not simply increase the number of cancers diagnosed but also improve the early detection of biologically relevant cancer [46]. Despite the small number of cancers in our study, we could postulate that DWI, by providing functional information within a tissue, may represent a valid screening tool for detecting clinically relevant cancers rather than lesions that would not cause harm during a lifetime.

In our study, breast DWI showed a sensitivity of 73% and a specificity of 90% for R1, while it showed a sensitivity of 64% and a specificity of 90% for R2, which is reasonable for the significantly less experience of R2 than R1, and these results are consistent with other studies [47,48]. For both readers, the specificity was high, according to a screening group with a much higher probability of true negatives, while the sensitivity was higher for the reader with more experience. All these data suggest that experience plays an important role. There is no reason to believe that reading FP-MR imaging as long as DWI images would require less expertise than reading screening mammograms. Radiologists are required to read a certain number of mammograms in order to participate in any mammographic screening campaigns. First, we should boot DWI training programs before using it in a screening setting.

The expert reader encountered three false negatives cases. One was a 6 mm tumor with low-grade histology, located in the axillary extension, representing a low-sensitivity location, as lymph nodes mimic the DWI behavior of cancer, and they are usually found in the axillary cavity and axillary extension. The FP-MRI was reported as probably benign (BI-RADS 3), as the nodule was oval-shaped with regular margins and homogeneous enhancement (Figure 2). The second false negative case was a 10 mm nodule with irregular margins and inhomogeneous contrast enhancement, albeit unchanged in size and morphology for at least 4 years, reported as benign (BI-RADS 2) even with the FP-MRI (Figure 3). The third false negative case was a 10 mm lesion with contrast enhancement in FP-MRI, but not detectable in DWI nor in T2-WI (Figure 4): the FP-MRI was reported as suspicious (BI-RADS 4), and the patient underwent biopsy with histological diagnosis of high-grade, poorly differentiated, triple-negative breast cancer. We hypothesize that this aggressive missed cancer was due to its peripheral location, small size, and the un-optimized protocol before the EUSOBI statement, published in 2020 [13].

The ACR benchmarks for screening MRI are sensitivity >80% and specificity >85–90% [17]. Vreemann et al. showed that the performance of screening is highly dependent on the actual screening indication, with a sensitivity of only 81% in BRCA1 carriers [49].

As our sensitivity was below the required benchmarks, we retrospectively calculated the sensitivity of our FP-MRI, obtaining a sensitivity of 73% and a slightly higher specificity (95% FP-MRI vs. 90% DWI protocol). All, except one, DWI-detected cancers were triple-negative cancers, and all lymph nodes were negative T1 tumors.

Our results are consistent with previous studies in which DW-MRI showed an overall lower sensitivity but higher specificity compared to DCE-MRI [15,44,45,50,51] (Table 5), and moderate agreement between readers with different levels of experience [47], emphasizing the importance of training and experience.

As awareness and concerns of gadolinium-containing contrast agents increase [52] and the emerging emphasis on patient-centered care reshape radiologist practice [53,54], the efforts for breast cancer screening using DW-MRI are gaining importance. Among our false negatives, only one was a triple-negative, poorly differentiated, high-grade, 9 mm cancer. DWI is already known to have low sensitivity in small invasive breast cancers, in some types of invasive lobular carcinomas, in some types of low-grade DCIS, and in mucinous breast cancer [38]. We can hypothesize that DWI may miss at least some of those biologically irrelevant cancers but much more rarely aggressive invasive breast cancers. In addition, in a wider population, using DW-MRI as a screening tool might reduce the overdiagnosis and descale treatment of certain DCIS.

DWI is not perfect, and the major drawback is the suboptimal spatial resolution, limiting morphologic analysis. In the future, the use of high-field strength scanners and the latest high-resolution DW-MRI sequences might improve the detection and characterization of breast lesions.

Furthermore, artificial intelligence algorithms applied to this protocol may, in the future, play an important role in breast cancer screening, improving its quality and assisting radiologists with the increased workload due to the expansion of screening indications [55,56].

Therefore, DWI supplemented with T2-WI could be considered as a possible screening tool in a high-risk population group. Accordingly, our results encourage prospective studies investigating the potential of unenhanced breast MRI as a screening tool.

Our study is limited by several factors beyond the retrospective design. First, it was performed at a single tertiary-level cancer care institution, with dedicated breast equipment and radiologists, so the results may not be generalizable to general practice. Second, the population was a small cohort, and the median lesion size was relatively small, which may have affected the sensitivity. Third, we evaluated the utility of DWI in women with BRCA1/2 mutations who are not representative of the general screening population. Finally, the examinations evaluated had different technical parameters since the DWI sequences changed during our evaluation period. The parameters were optimized in 2020 since the Institute adopted the minimum requirements for breast DWI according to EUSOBI guidelines [13].

## 5. Conclusions

In high-risk patients, DWI might represent a viable screening tool if confirmed by further prospective trials. Additionally, DWI might be implemented in a subgroup of high-risk patients unsuitable for contrast agent application or be complementary to mammography, even for different population groups regarding the risk. Breast DWI supplemented with T2-WI allowed breast cancer detection with good sensitivity and high specificity, comparable, therefore, to other screening tests, particularly for an expert reader. The results open the way to prospective studies investigating the potential role of DWI and T2-WI as a stand-alone method for unenhanced breast MRI population-wide screening.

## Figures and Tables

**Figure 1 diagnostics-13-01996-f001:**
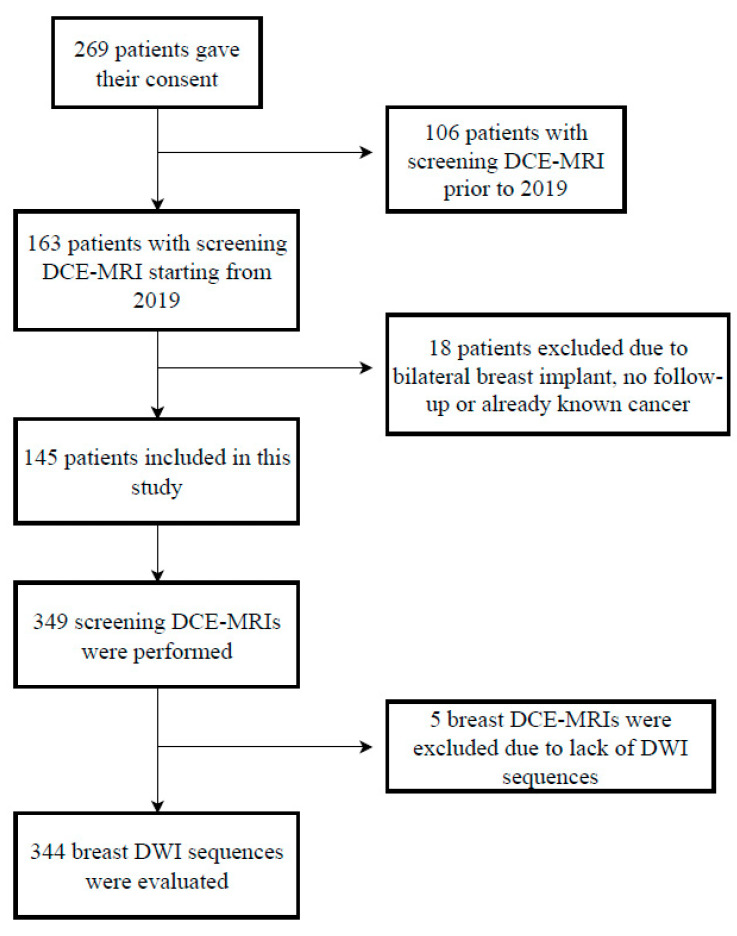
Flowchart of the study. DCE-MRI—dynamic contrast-enhanced magnetic resonance imaging.

**Figure 2 diagnostics-13-01996-f002:**
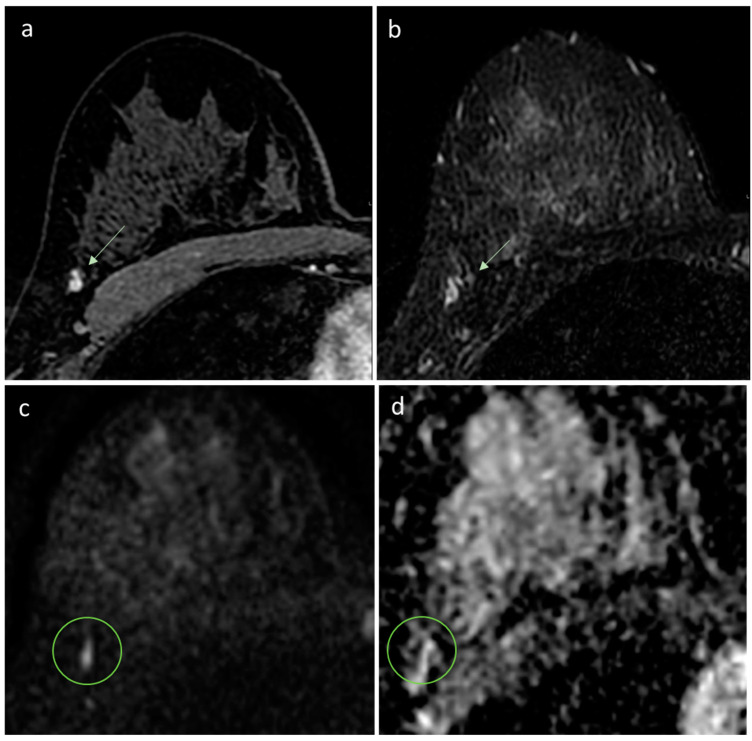
MRI of right breast cancer. (**a**) First dynamic postcontrast T1-weighted fat-saturated image reveals an oval mass (arrow) with well-circumscribed margins at the right axillary tail. (**b**) T2-weighted image shows a corresponding hyperintense regular mass, thus, considered benign (BI-RADS category 2 assessment). (**c**,**d**) The diffusion-weighted image shows a small hyperintensity mass (circle) with no discernible lesion on the ADC map. The woman underwent prophylactic mastectomy, and a grade 3 invasive ductal carcinoma was found in the right breast.

**Figure 3 diagnostics-13-01996-f003:**
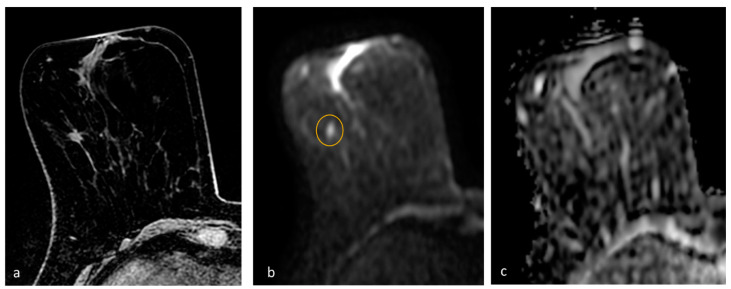
MRI of right breast cancer. (**a**) First dynamic postcontrast T1-weighted fat-saturated image reveals an irregular mass with spiculated margins and heterogeneous internal pattern in the right breast. FP-MRI shows long-term resonance stability, and mass was, thus, considered benign (BI-RADS category 2 assessment). (**b**,**c**) Diffusion-weighted image shows a small hyperintensity (circle) without a corresponding hypointensity on the ADC map. The woman underwent bilateral prophylactic mastectomy, which demonstrated a grade 1 invasive ductal carcinoma, estrogen- and progesterone-receptor-positive, with 10% proliferation index.

**Figure 4 diagnostics-13-01996-f004:**
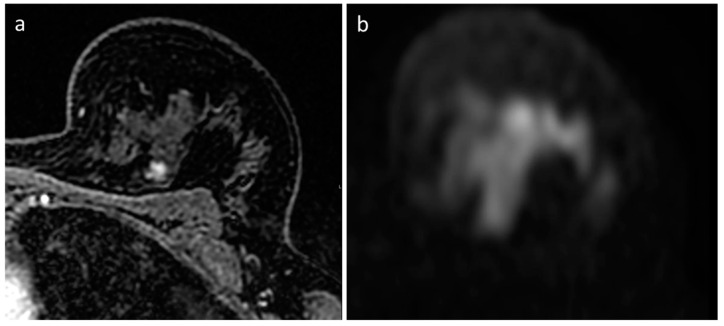
MRI of left breast cancer. (**a**) First dynamic postcontrast T1-weighted fat-saturated image shows a round mass with irregular margins and heterogeneous enhancement at the periphery of the inner upper quadrant of the left breast. (**b**) No discernible diffusion restriction lesion at the correlating site in the diffusion-weighted image. The woman underwent MRI-guided biopsy, which revealed an invasive ductal carcinoma. The final histology confirmed a grade 3 invasive ductal carcinoma, triple-negative, with 55% proliferative index.

**Table 1 diagnostics-13-01996-t001:** Inclusion and exclusion criteria. MRI: magnetic resonance imaging.

Inclusion Criteria:	Exclusion Criteria:
≥18 years old women	Pregnancy or breastfeeding
At least 1 year of clinical and radiological follow-up or histological analysis through biopsy or surgery.	No follow-up or no pathological gold standard by needle biopsy or surgery.
Written informed consent for MRI signed and dated by the patient and the radiologist prior to inclusion.	Patients undergoing neoadjuvant chemotherapy.
BRCA1 and BRCA2 mutation carriers.	Common contraindications to MRI (presence of MR-incompatible devices, history of severe claustrophobia, and side effects due to MRI contrast agents).
	Symptoms or signs of breast cancer or recurrence.
	Bilateral breast implants.

**Table 2 diagnostics-13-01996-t002:** Technical parameters for diffusion-weighted imaging of the breast (TE—echo time; TR—repetition time; EPI—echo-planar imaging).

Acquisition Parameter	Pre-DWI EUSOBI Consensus	Post-DWI EUSOBI Consensus
Type of sequence	EPI	EPI
Orientation	2D axial	2D axial
Field of view	The field of view covers both breasts	The field of view covers both breasts
In-plane resolution	2 × 3.6 mm^2^	2 × 2 mm^2^
Slice thickness	5.0 mm	3.5 mm
Spacing between slices	0.5 mm	0.4 mm
Number of b values	2	2
Lowest b value	0 s/mm^2^	0 s/mm^2^
High b value	800 s/mm^2^	800 s/mm^2^
Fat saturation	Yes	Yes
TE	Minimum possible	Minimum possible
TR	≥3000 ms	≥3000 ms
Acceleration	2	2
Post-processing	Generation of ADC maps	Generation of ADC maps

**Table 3 diagnostics-13-01996-t003:** Histology of cancers. * Maximal size dimension; ** year of diagnosis; ⬫ Ki-67 (marker ff proliferation). DIN—ductal intraepithelial neoplasia; IDC—invasive ductal carcinoma; TNBC—triple-negative breast cancer; ER—estrogen receptor; PR—progesterone receptor; HER2—human epidermal growth factor receptor 2.

Patients with Diagnosis of Breast Cancer (*n* 11)					
	Year **	Histology	Receptor Status	Size * (mm)	Ki-67 (%) ⬫
1	2021	DIN 2	ER+ = 95%; PR+ = 5%; HER2 weakly + in 60%	12	10%
2	2021	DIN 2	ER−; PR−; HER2 weakly + in 15%	20	28%
3	2019	IDC	TNBC	13	80%
4	2021	DIN 2	ER+ = 95%; PR+ = 90%; HER2 weakly + in 40%	20	18%
5	2019	Secretory carcinoma	TNBC	5	5%
6	2021	DIN 2		4	
7	2020	Poorly differentiated IDC	ER-; PR weakly +; HER2 weakly + in 40%	12	40%
8	2021	DIN 2	ER+ = 95%; PR+ = 5%; HER2 weakly + in 75%	5	14%
9	2020	IDC	TNBC	18	40%
10	2021	Well-differentiated IDC	ER+ = 95%; PR+ = 70%; HER2 neg	9	10%
11	2019	Moderately differentiated apocrine breast cancer	TNBC	8	23%

**Table 4 diagnostics-13-01996-t004:** Readers’ diagnostic performances. PPV—positive predictive value, NPV—negative predictive value.

R1				R2			
		95% CI				95% CI	
Sensitivity	0.73	0.46	0.99	Sensitivity	0.64	0.35	0.92
Specificity	0.90	0.87	0.94	Specificity	0.90	0.87	0.94
PPV	0.20	0.08	0.32	PPV	0.18	0.06	0.30
NPV	0.99	0.98	1.00	NPV	0.99	0.97	1.00
Accuracy	0.90			Accuracy	0.90		

**Table 5 diagnostics-13-01996-t005:** Readers’ diagnostic performances. PPV—positive predictive value, NPV—negative predictive value.

	Field Strength	Number of Cancer	Study Population	Sensitivity	Specificity
Baltzer at al. 2010 [44]	1.5 T	54	Consecutive BI-RADS 4 and 5 masses	94.4% (average of R1 and R2)	85.2%
Rotili et al. 2020 [15]	1.5 T	96	Consecutive mixed screening, staging, and follow-up	87%	90.5% (average of R1 and R2)
Baltzer at al. 2018 [45]	3.0 T	67	Consecutive with conventional imaging BI-RADS 3 and 4	91%	73.2%
Yamada et al. 2018 [50]	1.5 T	89	Consecutive with suspicious findings using conventional imaging	92.7% (average of R1 and R2)	95.8% (average of R1 and R2)
Bickelhaupt S et al. 2017 [51]	1.5 T	22	Consecutive with conventional imaging BI-RADS 4 and 5	90%	85.9%
Present study	1.5 T	11	BRCA1 and BRCA2 asymptomatic carriers	69% (average of R1 and R2)	90%

## Data Availability

The data presented in this study are available on request from the corresponding author. The data are not publicly available due to privacy reasons.

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
