# Peer review of "An Unenhanced Breast MRI Protocol Based on Diffusion-Weighted Imaging: A Retrospective Single-Center Study on High-Risk Population for Breast Cancer"

_diagnostics, 2023, doi:10.3390/diagnostics13121996_

Round 1

Author Response

Dear Reviewer,

We are grateful for your review and your valuable comments.

With the aim of improving the quality of our paper, according to your valuable suggestions and to comments of other reviewers, we have changed several parts of the manuscript.

Please find below how we revise the manuscript for each point you kindly pointed out:

  1. Title is not suitable according to the structure of the manuscript. Revise it accordingly.

The title of the paper was changed into: An un-enhanced breast MRI protocol based on diffusion-weighted imaging: a retrospective single center study in high-risk population for breast cancer.

  1. The abstract looks to brief, it is suggested to give more summary of the carried-out work, also the keywords are not mentioned, it should be 6 to 10.
  2. The novelty of the item should be clearly stated in the abstract section.

We changed the abstract according to you suggestion as follows:

PURPOSE: The study aimed to investigate the use of contrast-free magnetic resonance imaging (MRI) as an innovative screening method for detecting breast cancer in high-risk asymptomatic women. Spe-cifically, the researchers evaluated the diagnostic performance of diffusion-weighted imaging (DWI) in this population.

METHODS: MR images from asymptomatic women, carrier of a germline mutation in either the BRCA1 or BRCA2 gene collected in a single centre from January 2019 to December 2021 were retrospectively evaluated. A radiologist with experience of breast imaging (R1) and a radiology resident (R2) independently evaluated DWI/ADC map and, in case of doubts, T2-WI. The standard of reference was the pathological diagnosis through biopsy or surgery, or ≥ 1 year of clinical and radiological follow-up. Diagnostic performances were calculated for both readers with a 95% confidence interval (CI). The agreement was assessed using Cohen’s kappa (κ) statistics.

RESULTS: Out of 313 women, 145 women were included (49.5 ± 12 years), totaling 344 breast MRI with DWI/ADC map. Per-exam cancer prevalence was 11/344 (3.2%). Sensitivity was 8/11 (73%; 95% CI: 46-99%) for R1 and 7/11 (64%; 95% CI: 35-92%) for R2. Specificity was 301/333 (90%; 95% CI: 87-94%) for both readers. Diagnostic accuracy was 90% for both readers. R1 recalled 40/344 exams (11.6%) and R2 recalled 39/344 exams (11.3%). Inter-reader reproducibility between readers was moderate agreement (κ = 0.43).

CONCLUSIONS: In women carrier of a BRCA1/2 mutation, breast DWI supplemented with T2-WI allowed breast cancer detection with high sensitivity and specificity by a radiologist with high experience of breast imaging, comparable to other screening tests. The findings suggest that DWI and T2-WI have the potential to serve as a stand-alone method for un-enhanced breast MRI screening in a selected population, opening up new perspectives for prospective trials.

Moreover, we included the following keywords: Breast neoplasm; Radiology; Magnetic Resonance Imaging; Diffusion Weighted Imaging; Prevention; Personalized Medicine.

  1. Write the motivation and contribution of your study in the introduction section (a short paragraph)

Thanks for your suggestion. We included the following paragraph into the introduction section:

(…) Accordingly, the motivation behind this study is to investigate the potential of a contrast-free MRI technique using DWI supplemented with T2-WI for breast cancer screening in women at high-risk of developing breast cancer due to a germline mutation in either the BRCA1 or BRCA2 gene, aiming to open new perspectives for prospective trials investigating the potential role of un-enhanced breast MRI for screening in a selected population, potentially leading to improving early detection and outcomes with a more cost-effective screening method for women at high-risk of developing breast cancer.

  1. Authors have not written the organization of the remaining part of manuscript section.
  2. There is missing of related work section after the introduction section. Summarize some relevant and latest survey articles and compare your contributions with the existing surveys.

We updated and completed the missing works into the entire manuscript. However, we do not are afraid to understand completely which survey we could consider and compare.

  1. In Materials and Method section in Table 3, authors have taken data for only year 2021, but not for the year 2022 & 2023.

As our population included asymptomatic women with BRCA1/2 mutations who underwent screening breast MRI, namely full protocol MRI (FP-MRI), from January 2019 to December 2021 in our Institute – as we reported in materials and methods section - we believe that data of the year 2022 and 2023 should not be considered for this study.

  1. The manuscript should be refined for English grammatical structure and phraseology. The manuscript should be polished by an English linguist.

Thanks for your suggestion. Accordingly, we subjected the entire manuscript to an editing review by a native English speaker.

  1. Authors should declare the acronyms first whenever they appear in the literature first. 10. Overall work is average but the organization/presentation of the paper is ordinary. Writing and references need correction and there are many flaws in write-up and need to be removed. 11. The work still needs to be refined and presentation of the work also needs to be improved. 12. Proposed methodology need to be elaborated with more detail and how it is efficient must be stated. 13. The proposed method lacks motivation and justifications. More advantages should be discussed in detail. 14. The novelty is limited. What is the contribution of this paper from the modelling perspective? 15. Authors also must provide some insight discussion of the results. 16. What is the complexity of proposed approach? What the authenticity/correctness of proposed protocol? 17. It looks a comprehensive literature survey is missing in the article, authors are suggested to add more latest references, some suggestions are given below for the consideration. https://www.sciencedirect.com/science/article/pii/S1877050922011814 https://www.sciencedirect.com/science/article/pii/S1047320322000396 https://www.mdpi.com/2071-1050/14/21/13998

https://www.sciencedirect.com/science/article/pii/S0030402622017235 

We are thankful again for your comments. We have modified our study to improve the overall quality, especially providing some insights discussion of the results, and highlighting the innovation of our research. Again, we are a little bit confused about the need to include a literature survey analysis into the paper, as it seems to be out of the topic of our study, like the references suggest appears too.

Finally, you can find the reviewed manuscript in both the tracked- and the clean-version in the attachments. We hope to have appropriately answered your concerns and that the paper revised following your comments finds your standard of quality and it is worthy of publication in Diagnostics journal. 

Reviewer 2 Report

The manuscript aims to evaluate contract-free magnetic resonance imaging as a screening method to detect breast cancer. The authors claimed that the diagnostic performance of DWI in asymptomatic women with high-risk breast cancer opened new areas of prospective trials investigating DWI's role in selected populations. Although the technique is not general and highly accurate, at least it paves the path for more research on the same topic. I believe the manuscript is suitable for publication in the journal "Diagnostics" only after solving the significant issues.

Major points:

1.      The introduction section (except for a few sentences) was directly copied from different sources. You are not allowed to do that. Please revise as necessary.      

2.      In line 62, the authors should use references when introducing DWI.

3.      Use reference for table 1 as it has been copied from the author's earlier publication in Cancers journal. Also, lines 101-109 are copied.

4.      Can the author provide a comparative table consisting of performance metrics between their experiment and references 43,44,45?

5.      Another table consists of the comment made in lines 315-316.

6.      What is the way forward against the limitation provided in lines 340-348?

7.      Also, given that those are significant limitations against making the author's technique generalized, what modification can be done to make it generalized and more accurate?

There have been significant overlaps and direct copies throughout the whole manuscript, which came from different sources. Authors should refrain from such. In the revised submission, authors are suggested to take care of these problems carefully.

Author Response

The manuscript aims to evaluate contract-free magnetic resonance imaging as a screening method to detect breast cancer. The authors claimed that the diagnostic performance of DWI in asymptomatic women with high-risk breast cancer opened new areas of prospective trials investigating DWI's role in selected populations. Although the technique is not general and highly accurate, at least it paves the path for more research on the same topic. I believe the manuscript is suitable for publication in the journal "Diagnostics" only after solving the significant issues.

Dear Reviewer,

Many thanks for your time for reviewing our paper, for your valuable comments and for your overall positive evaluation.

You can find in the attachments the reviewed manuscript in both the tracked- and the clean-version and below the responses for each of the issues you rightly pointed out.

Major points:

  1. The introduction section (except for a few sentences) was directly copied from different sources. You are not allowed to do that. Please revise as necessary.      

The introduction was rewritten entirely.

  1. In line 62, the authors should use references when introducing DWI.

References were included accordingly.

  1. Use reference for table 1 as it has been copied from the author's earlier publication in Cancers journal. Also, lines 101-109 are copied.

We slightly modified table 1 and the reported lines. However, we believe that a reference for an inclusion/exclusion criteria table would be inappropriate. The criteria are the same of the other study you mentioned that we published in  Cancers: this is the reason why it appears the same.

  1. Can the author provide a comparative table consisting of performance metrics between their experiment and references 43,44,45?
  2. Another table consists of the comment made in lines 315-316.

Regarding these points 4 and 5, we provide a new Table (namely, Table 5) consisting of performance metrics with our results and previous studies, cited in the bibliography (we remove the reference 44, namely the study by Moran et al., as it was inappropriate in the context: sorry for this typo!). Thanks also for this comment, we believe that this table will improve the overall quality of our paper.

  1. What is the way forward against the limitation provided in lines 340-348?

We added some possible ways forward to overcome the limitations of our study in the discussion section.

There have been significant overlaps and direct copies throughout the whole manuscript, which came from different sources. Authors should refrain from such. In the revised submission, authors are suggested to take care of these problems carefully.

We revised the whole manuscript in order to eliminate the overlaps with others similar articles.

We hope to have appropriately replied to your concerns and that the reviewed paper finds your standard of quality and it is now worthy of publication in Diagnostics Journal. 

Reviewer 3 Report

Thank you for requesting  to provide a review of this revised article, about the un-enhanced breast MRI protocol based on duffusion-wighted imaging.

   The main purpose of the analysis was to assess the potential of a contrast-free MRI technique by using DWI supplement with T2-WI for breast cancer screening in common at high-risk of developing breast cancer. The main question adressed in the research was whether DWI images of asymptomatic high-risk women undergoing MRI with a screening indication can be used to accurately determine breast cancer and to assess lesion visibility of clinically significant cancer. In the article, the diagnostic performance of DWI sequences were tested, which were supplemented with T2-weighted sequence, as a tool for cancer detection in asymptomatic BRCA carriers undergoing annual screening MRI.

   The study was a retrospective study, conducted of women with BRCA1/BRCA2 mutations who underwent screening breast MRI for a period of time between Januay 2019 and December 2021. The topic is original and relevant in the field and brings usefull knowledge regarding the subject. A comprehensive search strategy was used. The review methodology was comprehensive with screening and data extraction. When it comes to the methodology used, no specific improvements should be considered from my point of view.

   The conclusions are consistent with the evidence and the arguments presented, and they adress properly to the main question which conducted the analysis.

   The references have been verified and are appropriate for the study. 

    Regarding the figures and pictures used in the article, they are very understandable and easy to be followed and they adress properly for this kind of study, so no other comments regarding this subject are necessary. Also, the pictures with the MRI examinations that have been done to the patients, were very explicit. 

  Regarding the structure and accuracy of the phrases, the manuscript has well structured information, with supported evidence and well structured phrases.

   The manuscript is original and well defined. The results provide an advance in current knowledge. The results are being interpreted appropriately and are significant, as well as the conclusions.

  The study is correctly designed and the analysis is being performed at high standards, so the data are robust enough to draw the conclusion. Surely the paper will attract a wide readership. 

   To conclude, the article is written in a proper way and brings useful information regarding the subject. 

   However, I still have a few comments to add in the lines below.

Line 45: is one of the leading cause, not „to be one of the leading cause”

Line 54: displayed, not „displaying”

Line 58: for screening, not „to screen”

Line 66: selected, not „select”

Line 106: regarding the diagnostic, not „about diagnostic”

Line 139: included, not „includes”

Line 262: which includes, not „which include”

Line 338: before use it as a screening setting, not „before its a in a screening setting”

Line 341: behaviour, not „behavior”

Author Response

Thank you for requesting  to provide a review of this revised article, about the un-enhanced breast MRI protocol based on duffusion-wighted imaging. 

   The main purpose of the analysis was to assess the potential of a contrast-free MRI technique by using DWI supplement with T2-WI for breast cancer screening in common at high-risk of developing breast cancer. The main question adressed in the research was whether DWI images of asymptomatic high-risk women undergoing MRI with a screening indication can be used to accurately determine breast cancer and to assess lesion visibility of clinically significant cancer. In the article, the diagnostic performance of DWI sequences were tested, which were supplemented with T2-weighted sequence, as a tool for cancer detection in asymptomatic BRCA carriers undergoing annual screening MRI. 

   The study was a retrospective study, conducted of women with BRCA1/BRCA2 mutations who underwent screening breast MRI for a period of time between Januay 2019 and December 2021. The topic is original and relevant in the field and brings usefull knowledge regarding the subject. A comprehensive search strategy was used. The review methodology was comprehensive with screening and data extraction. When it comes to the methodology used, no specific improvements should be considered from my point of view. 

   The conclusions are consistent with the evidence and the arguments presented, and they adress properly to the main question which conducted the analysis. 

   The references have been verified and are appropriate for the study.  

    Regarding the figures and pictures used in the article, they are very understandable and easy to be followed and they adress properly for this kind of study, so no other comments regarding this subject are necessary. Also, the pictures with the MRI examinations that have been done to the patients, were very explicit.  

  Regarding the structure and accuracy of the phrases, the manuscript has well structured information, with supported evidence and well structured phrases. 

   The manuscript is original and well defined. The results provide an advance in current knowledge. The results are being interpreted appropriately and are significant, as well as the conclusions. 

  The study is correctly designed and the analysis is being performed at high standards, so the data are robust enough to draw the conclusion. Surely the paper will attract a wide readership.  

   To conclude, the article is written in a proper way and brings useful information regarding the subject.  

   However, I still have a few comments to add in the lines below. 

Line 45: is one of the leading cause, not „to be one of the leading cause”

Line 54: displayed, not „displaying” 

Line 58: for screening, not „to screen” 

Line 66: selected, not „select” 

Line 106: regarding the diagnostic, not „about diagnostic” 

Line 139: included, not „includes” 

Line 262: which includes, not „which include” 

Line 338: before use it as a screening setting, not „before its a in a screening setting” 

Line 341: behaviour, not „behavior” 

Dear Reviewer,

Thank you for taking the time to review the revised article on the un-enhanced breast MRI protocol based on diffusion-weighted imaging and for your overall positive evaluation.

Particularly, we are glad to read that you found the article to be well-structured, with supported evidence and well-structured phrases. Your feedback on the comprehensiveness of the search strategy and the data extraction methodology is appreciated.

We are pleased that you found the study to be original and relevant, with the potential to bring useful knowledge to the field. Your positive comments on the diagnostic performance of DWI sequences supplemented with T2-weighted sequence for cancer detection in asymptomatic BRCA carriers undergoing annual screening MRI, and the accuracy and robustness of the analysis, are encouraging. We also appreciate your feedback on the clarity and usefulness of the figures and pictures used in the article.

You can find in the attachments the reviewed manuscript in both the tracked- and the clean-version with the correction you rightly pointed out.

We hope to have appropriately replied to your concerns and that the reviewed paper finds your standard of quality and it is now worthy of publication in Diagnostics Journal. 

Round 2

Reviewer 1 Report

All the comments are not addressed by the authors given in the Round 1. So revise it carefully according to the previous comments.

Author Response

All the comments are not addressed by the authors given in the Round 1. So revise it carefully according to the previous comments. 

Dear Reviewer,

We are grateful for your further review and your time.

In this latest revision, we have made every effort to address all of the issues you have raised, and we hope that you will find the revised version of our article satisfactory. We would like to thank you for your continued support and encouragement throughout this process.

We do have some concerns, however, about your comment regarding a comprehensive literature survey and the inclusion of papers that may not be related to the topic of our study. While we appreciate your suggestion, we are unsure how including papers on deep learning and machine learning techniques would be relevant to our research on the un-enhanced breast MRI protocol based on diffusion-weighted imaging. We would be grateful if you could provide us with further clarification on this matter.

Apart for that, we tried to solve all the other comments previously remained unsolved as you can check on the attached papers.

Moreover, we would like to take this opportunity to inform you that, in addition to addressing all of the concerns you raised in your review, we have also taken steps to further improve the quality of our manuscript. Specifically, we have asked a native English speaker to review our paper and suggest any necessary improvements to ensure that the language is clear and concise.

We are committed to producing a manuscript that meets the highest standards of quality, and we believe that this additional step will help us achieve that goal. We appreciate your understanding and support as we strive to make our research as accessible and impactful as possible.

As in our previous correspondence, we have included both the tracked- and clean-versions of the manuscript for your convenience.

We sincerely hope that we have addressed all of your concerns and that our revised paper meets the high standards of quality required for publication in Diagnostics journal.

Reviewer 2 Report

The authors have made substantial changes and answered all the queries. Here are some minor issues that need their attention.

1.      Line 147-154 needs to change significantly. It has been taken from another MDPI publication. If authors have used the same technique in both places, they are supposed to cite the previous source.

2.      Why is data from 2022 not present in table 3?

3.      What is the explanation for poor sensitivity in R1 and R2 for the reader's diagnostic performance? The authors mentioned it is consistent with other studies, but those studies involve some familiar authors. Can they explain the metric's value using other's studies?

Author Response

The authors have made substantial changes and answered all the queries. Here are some minor issues that need their attention. 

Dear reviewer, 

many thanks for your consideration and your time for this second round of review of our article. 

  1. Line 147-154 needs to change significantly. It has been taken from another MDPI publication. If authors have used the same technique in both places, they are supposed to cite the previous source. 

 Thanks for your comment. We changed that section accordingly as follows:

The exams were evaluated and scored based on the BI-RADS diagnostic classification. However, in the final unenhanced assessment, the study did not permit the use of the BI-RADS 0 category, and BI-RADS 6 was not possible due to blinded reading, which was intended to mimic a screening scenario. As a result, the scale was divided into two categories: negative (BI-RADS 1, 2) and positive (BI-RADS 3, 4, and 5), consistent with standard screening reading. The recall examinations involved measuring the largest diameter of the main lesion.

  1. Why is data from 2022 not present in table 3? 

This is because the our population included asymptomatic women with BRCA1/2 mutations who underwent screening breast MRI from January 2019 to December 2021 in our Institute, as we reported in materials and methods.

  1. What is the explanation for poor sensitivity in R1 and R2 for the reader's diagnostic performance? The authors mentioned it is consistent with other studies, but those studies involve some familiar authors. Can they explain the metric's value using other's studies? 

The poor sensitivity in R1 and R2 for the reader's diagnostic performance could be due to a variety of factors. One possibility is that unenhanced breast MRI may be more difficult to interpret than contrast-enhanced MRI or other imaging modalities (especially for low experience readers), as it can be challenging to differentiate between normal breast tissue and potential lesions. Additionally, reader factors such training and interpretation skills can also play a role in diagnostic accuracy. The specific metric that has been used was the true positive rate, which measures the proportion of actual positive cases that are correctly identified by the reader. A low TPR indicates a high rate of false negatives, or cases that are incorrectly classified as negative when they are actually positive. This metric is commonly used in medical imaging studies to evaluate the diagnostic accuracy of different readers or imaging modalities.

Round 3

Reviewer 1 Report

It can be accepted.